# Gate-tunable negative longitudinal magnetoresistance in the predicted type-II Weyl semimetal WTe$_2$

Yaojia Wang[1], Erfu Liu[1], Huimei Liu[1], Yiming Pan[1], Longqiang Zhang[1], Junwen Zeng[1], Yajun Fu[1], Miao Wang[1], Kang Xu[1], Zhong Huang[1], Zhenlin Wang[1], Hai-Zhou Lu[2], Dingyu Xing[1], Baigeng Wang[1], Xiangang Wan[1] & Feng Miao[1]

The progress in exploiting new electronic materials has been a major driving force in solid-state physics. As a new state of matter, a Weyl semimetal (WSM), in particular a type-II WSM, hosts Weyl fermions as emergent quasiparticles and may harbour novel electrical transport properties. Nevertheless, such a type-II WSM material has not been experimentally observed. In this work, by performing systematic magneto-transport studies on thin films of a predicted material candidate WTe$_2$, we observe notable negative longitudinal magnetoresistance, which can be attributed to the chiral anomaly in WSM. This phenomenon also exhibits strong planar orientation dependence with the absence along the tungsten chains, consistent with the distinctive feature of a type-II WSM. By applying a gate voltage, we demonstrate that the Fermi energy can be *in-situ* tuned through the Weyl points via the electric field effect. Our results may open opportunities for implementing new electronic applications, such as field-effect chiral devices.

[1] National Laboratory of Solid State Microstructures, School of Physics, Collaborative Innovation Center of Advanced Microstructures, Nanjing University, Nanjing 210093, China. [2] Department of Physics, South University of Science and Technology of China, Shenzhen 518055, China. Correspondence and requests for materials should be addressed to F. M. (email: miao@nju.edu.cn) or to X.W. (email: xgwan@nju.edu.cn) or to B.W. (email: bgwang@nju.edu.cn).

Since the discovery of topological insulators, which significantly enriched band theory[1,2], the possibility of realizing new topological states in materials other than insulators, such as semimetals or metals, has attracted substantial attention[3–7]. Weyl semimetals (WSMs), which host Weyl fermions[8] as emergent quasiparticles, have recently sparked intense research interest in condensed matter physics[3,9–16]. In WSMs, the conduction and valence bands linearly disperse across pairs of unremovable discrete points (Weyl points) along all three momentum directions[3,17], with the existence of Fermi Arc surface states as a consequence of separated Weyl points with opposite chirality[3]. Since the first theoretical prediction in pyrochlore iridates[3], several materials that break either the time-reversal or spatial-inversion symmetry have been proposed as WSMs, including a series of transition metal mono-phosphides[12,13]. These theoretical predictions have been experimentally confirmed by the observation of bulk Weyl points and surface Fermi Arcs[18–21], or the signature of chiral anomaly[15,22–30] via electric transport studies. Many other new properties, such as the topological Hall effect[14] and non-local quantum oscillations[31], have also been proposed.

The type-II WSM was recently proposed as a new type of WSM with Weyl points appearing at the boundary of electron and hole pockets[32–36]. Its distinctive feature of an open Fermi surface (in sharp contrast with a closed point-like Fermi surface in type-I WSMs) can induce exotic properties, such as planar orientation-dependent chiral anomaly. However, such type-II WSM materials have not been experimentally observed. As a unique layered transition-metal dichalcogenide that exhibits large and unsaturated (perpendicular) magnetoresistance (MR)[37], tungsten ditelluride (WTe$_2$) has been reported as a major material candidate for type-II WSM. While angle-resolved photoemission spectroscopy measurements encounter certain challenges in observing the Weyl points because of the limited experimental spectroscopic resolution[32,38], exploring the potential unique transport properties and realizing their tunability for future device applications are highly desirable.

In this report, low-temperature transport studies on thin WTe$_2$ samples are performed, revealing a clear negative longitudinal MR when the electric and magnetic fields are parallel. This phenomenon is highly angle sensitive and is suppressed by a small angle between the electric and magnetic fields, and this behaviour can probably be attributed to the chiral anomaly in the WSM. A unique property of type-II WSM, the planar orientation dependence, is also confirmed by the observed absence of negative longitudinal MR for all studied devices along the tungsten chains ($a$ axis). We further demonstrate that by applying a gate voltage, the Fermi energy of such a material can be effectively tuned through the Weyl points; thus, the unique transport properties can be controlled, suggesting possible applications in future chiral electronics.

## Results

**The selection and fabrication of thin-film devices.** WTe$_2$ is a T$_d$ type of transition-metal dichalcogenide (space group $Pnm2_1$) with a tungsten chain along the $a$ axis, as shown in Fig. 1a. The other principle axis, the $b$ axis, is perpendicular to the $a$ axis[37,39]. This T$_d$ phase breaks the inversion symmetry and was predicted to support the existence of type-II Weyl points[32,33]. We first focus on a key signature of the possibly existed Weyl points: the chiral-anomaly-induced negative longitudinal MR phenomenon. To make such observation feasible, thin flakes are required to sufficiently suppress the contribution of the strong positive longitudinal MR[40]. However, these thin flakes must be sufficiently thick, with energy bands similar to those of bulk crystals

(see Supplementary Fig. 1) to allow the existence of Weyl points. Thus, we selected thin WTe$_2$ flakes with thicknesses of 7–15 nm, which were prepared using the standard mechanical exfoliation method on a SiO$_2$ substrate and measured using an atomic force microscope. The crystalline orientations were identified using polarized Raman spectra[41] (see Supplementary Fig. 2).

Thin WTe$_2$ devices with metal electrodes were fabricated using a home-made shadow mask method[42], which effectively avoided undesirable wet process-induced doping in the pristine WTe$_2$ flakes[43]. A typical optical image of a four-probe device is shown in Fig. 1b, where the determined thickness of the thin flake was ~14 nm (inset of Fig. 1b). Figure 1c shows the schematic drawing of the device structure and four-probe MR measurement setup. Here, the angle between the applied magnetic field **B** and current direction **I** is $\theta$.

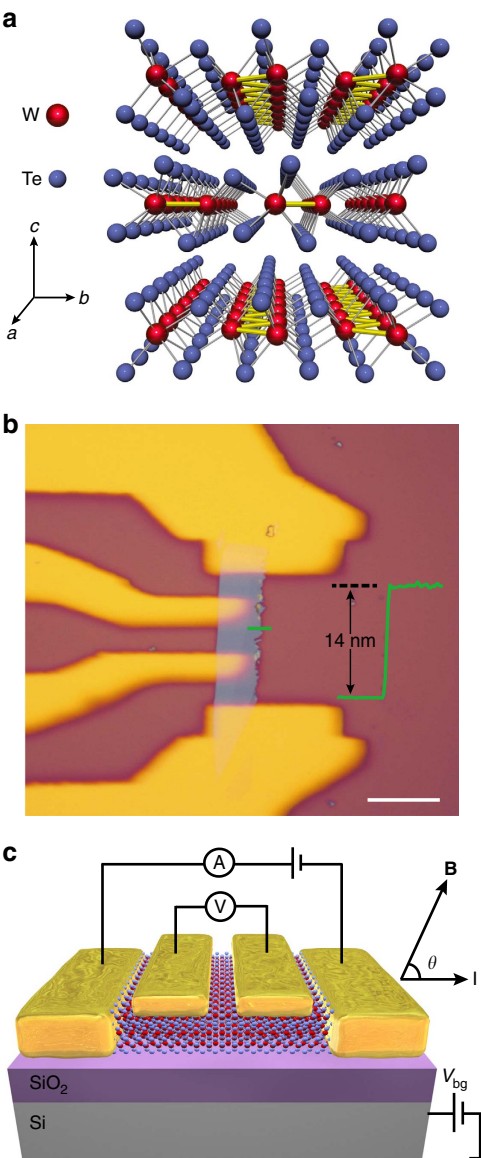

**Figure 1 | Thin WTe$_2$ film devices.** (**a**) The crystal structure of WTe$_2$; the yellow zigzag lines represent the tungsten chains along the $a$ axis. (**b**) Optical image of a four-probe thin WTe$_2$ film device. Scale bar, 15 μm. Inset: atomic force microscope (AFM) height profile of the flake along the green line. (**c**) Schematic structure and measurement circuit of the gated four-probe devices. The angle between the magnetic field and current is defined as $\theta$.

**Angle-sensitive negative longitudinal MR**. To examine the possible signal of the chiral anomaly, we performed longitudinal MR measurements on the devices by applying a magnetic field (from −12 to 12 T) parallel or at small angles to the current direction at 1.6 K. We observe two types of negative longitudinal MR phenomena when **B**//**I** ($\theta = 0°$), with typical data shown in Fig. 2a (sample #1) and Fig. 2b (sample #2). Both types of negative longitudinal MR exhibit strong angle sensitivity with the strongest signal at $\theta = 0°$ and an apparently suppressed signal at small $\theta$ when the magnetic field was slightly rotated (pronounced suppression at ~3.05° and −1.75° for samples #1 and #2, respectively). Within a relatively small range of the magnetic field, weak anti-localization (WAL) effect was observed and could be induced by the spin–orbit coupling in $WTe_2$ (ref. 44). Sample #1 shows only negative longitudinal MR at high magnetic field and the MR begins to decrease at approximately ±3.5 T and continues over the entire studied magnetic field range (until ±12 T). Sample #2 shows negative longitudinal MR with a positive MR signal at higher magnetic fields; the MR begins to decrease at approximately ±1.1 T and subsequently increases from approximately ±4.7 T. The observed positive longitudinal MR at higher magnetic fields is similar to what has been observed in TaAs[26,27] and TaP[45,46]. Its physical mechanism is still not clear, even though there are some theoretical proposals such as the Coulomb interactions among the electrons occupying the chiral states[26] or the anisotropy of the Fermi surface[47]. In our thin-flake samples, the positive longitudinal MR is much suppressed compared with the reported value (1,200%) in bulk crystals[40], making the observation of the negative longitudinal MR feasible. To fully understand why the positive longitudinal MR gets suppressed for thinner samples is theoretically challenging at current stage and requires more future research efforts.

While the negative longitudinal MR is rare in non-ferromagnetic materials, it can serve as one of the key transport signatures in WSMs. As the coupled Weyl points have opposite chiralities, the electrons are pumped from one point to the other and lead to a non-zero potential among them if the dot product of the magnetic and electric fields is not 0, that is, $\mathbf{B} \cdot \mathbf{E} \neq 0$. This chiral imbalance-induced potential will induce positive contribution to the conductance. Under the semi-classical approximation, when **B**//**E**, the anomaly conductivity[23] is described by

$$\sigma = \frac{e^4 v_F^3 \tau B^2}{4\pi^2 \hbar \Delta E^2} \qquad (1)$$

where $e$ is the electron charge, $v_F$ is the Fermi velocity near the Weyl points, $\Delta E$ is the measured chemical potential from the energy of the Weyl points and $\tau$ is the inter valley scattering time. The quadratic relation with a magnetic field leads to a negative MR effect with high sensitivity to the angle between **B** and **E**, which is consistent with our observations in thin $WTe_2$ samples.

There are few other origins other than the chiral anomaly, such as current jetting[48] and magnetic effects[49], which could induce the negative longitudinal MR effect under certain conditions. As $WTe_2$ is not a magnetic material, the possible origin of magnetic effects can be safely excluded. The current-jetting effect is usually induced by inhomogeneous currents generated when attaching point contact electrodes to a large bulk crystal. In our thin-film devices (rather than bulk crystals) with well-defined electrodes, it can be excluded as well[50]. Several theoretic proposals related to defects or impurities are also not applicable in our systems. For example, the negative longitudinal MR observed in our samples is not as temperature sensitive as the WAL effect (see Supplementary Fig. 3), suggesting it is not related to the defect-induced weak localization effect. Another theoretical work[51,52] predicting that certain impurities could induce negative longitudinal MR at small magnetic fields can be

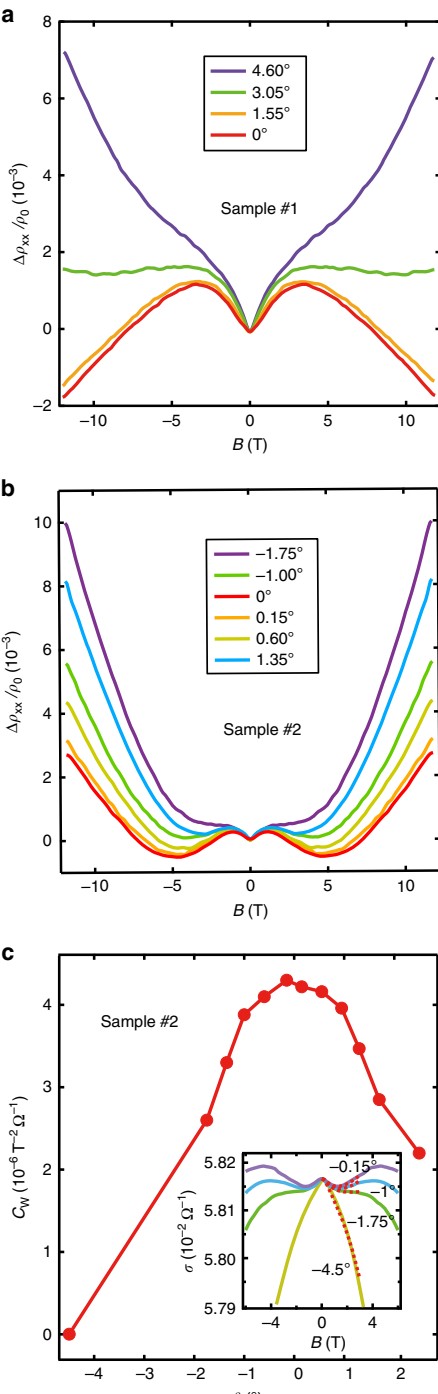

**Figure 2 | Angle-dependent negative longitudinal MR of thin $WTe_2$.** (**a**) Sample #1 exhibits only negative longitudinal MR at high magnetic fields, which is apparently suppressed at ~3.05°. (**b**) Sample #2 exhibits a negative longitudinal MR and a positive MR signal at higher magnetic field, which is apparently suppressed at approximately −1.75°. (**c**) The extracted chiral anomaly coefficient $C_W$ for sample #2 was obtained from fittings with the semi-classical formula. The results show strong angle $\theta$ sensitivity. Inset: fitting result (red dashed lines) of experimental magneto-conductivity curves (solid lines) at various angles. The MR data were collected at 1.6 K.

excluded, owing to the fact that our observations happen at much higher fields (up to 12 T). In the case of ultra-quantum limit, the impurities were also suggested to induce negative longitudinal MR in any three-dimensional metal, regardless of its

band structures[53]. To investigate this prediction, by analysing the measured Shubnikov–de Haas oscillations (see Supplementary Fig. 4 and Supplementary Table 1), we carefully calculated the Landau level indexes of different samples exhibiting negative longitudinal MR. The results indicate that the samples remain in the semi-classical limit.

Thus, the negative longitudinal MR can be quantitatively analysed using the formula[27,54] in the semi-classical limit, which includes the chiral anomaly contribution of the Weyl points:

$$\sigma_{xx}(B) = C_W B_{\parallel}^2 - C_{WAL}\left(\sqrt{B}\frac{B^2}{B^2+B_c^2} + \gamma B^2 \frac{B_c^2}{B^2+B_c^2}\right) + \sigma_0$$

(2)

where $C_W$ is the chiral coefficient, $C_{WAL}$ is the WAL coefficient, $B_c$ is the crossover critical field of two regions with different dependences (low field with $B^2$ dependence and higher field with $\sqrt{B}$ dependence)[54] and $\sigma_0$ is the zero field conductivity when **B//I**. For small $\theta$ values, the term of $\sigma_0$ is replaced by $\sigma_0/(1+\mu^2 B_{\perp}^2)$ to represent the contribution of transverse positive MR, where $\mu$ is the mobility. We analysed the angle-dependent longitudinal MR data of sample #2 when $0\,T < B < 3\,T$ and extracted the chiral coefficient $C_W$ from the fitting

results. The inset of Fig. 2c shows the fitting results of magneto-conductivity curves at various angles. The extracted $C_W$ versus $\theta$ data are plotted in Fig. 2c, revealing that $C_W$ is an effective parameter characterizing the strength of the contribution from chiral anomaly, which exhibits strong longitudinal angle sensitivity.

**The anisotropy of negative longitudinal MR.** A unique feature of the chiral anomaly in a type-II WSM is the predicted planar orientation dependence of the negative longitudinal MR effect due to the tilted band structure and coexistence of electron and hole pockets. We further examined the crystalline orientation dependence of the longitudinal MR along two principle axes. When the current was applied parallel to the $b$ axis (vertical to the tungsten chains), we observed negative longitudinal MR in all four measured samples (with aforementioned suitable thicknesses of 7–15 nm) (see Supplementary Fig. 5). In sharp contrast, for all four studied samples under similar conditions (exfoliated from the same batch of single crystals and in the identical thickness range) but different current orientation parallel to the $a$ axis (the tungsten chains), only positive longitudinal MR was observed (see Supplementary Fig. 6). These findings support the predicted

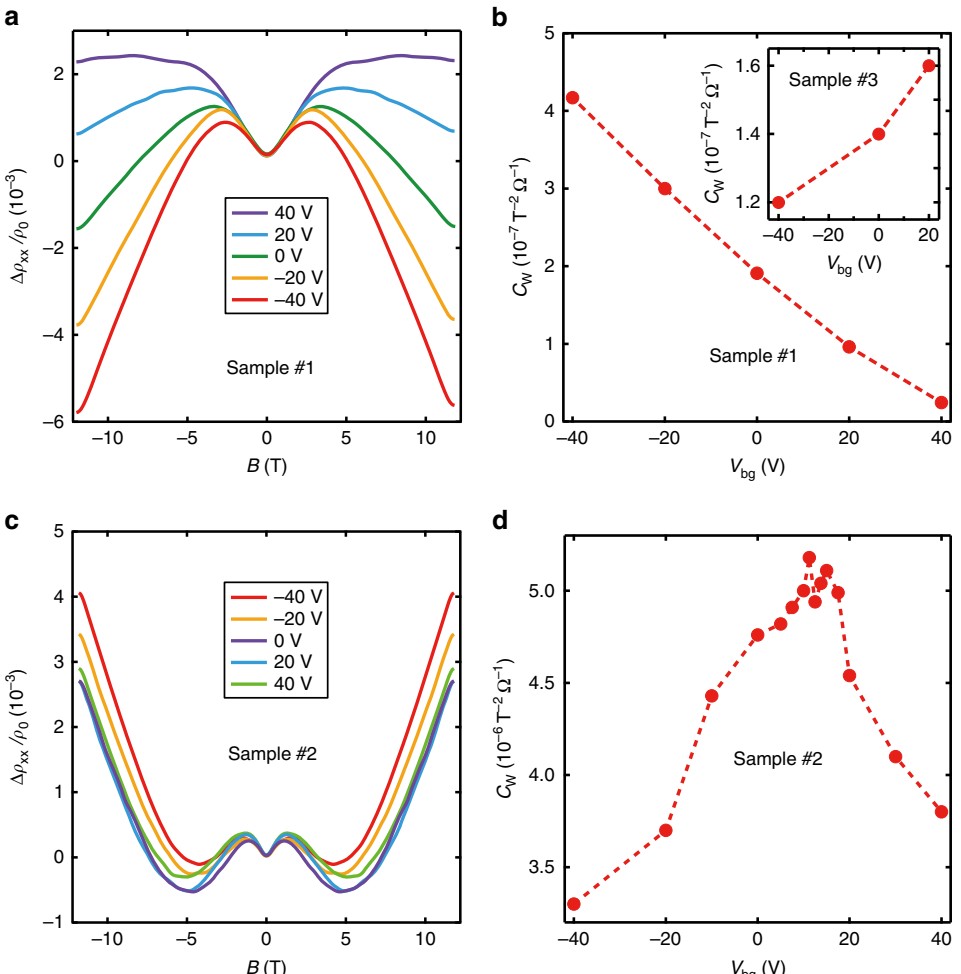

**Figure 3 | Gate-tunable negative longitudinal MR of thin WTe₂.** (**a**) The negative longitudinal MR of sample #1 for various $V_{bg}$, which shows a suppressed negative longitudinal MR effect with increasing $V_{bg}$ from $-40$ to $+40\,V$. (**b**) Plot of the extracted chiral anomaly coefficient $C_W$ of samples #1 (main) and #3 (inset), showing monotonous decreased/increased $C_W$ with increasing $V_{bg}$. (**c**) The negative longitudinal MR of sample #2 for various $V_{bg}$ shows a non-monotonous $C_W - V_{bg}$ dependence with a maximum $C_W$ at certain $V_{bg}$. (**d**) The $C_W$ data extracted from the dataset, where the maximum value of $C_W$ occurs at 10–17.5 V.

signature of the type-II Weyl fermion chiral anomaly in thin $WTe_2$ films (theoretical calculation in Methods).

**In-situ tuning of the Fermi energy through the Weyl points.** Compared with other experimentally studied WSMs (all bulk materials with fixed doping), another enormous advantage of a thin-layered type-II WSM is the potential for realizing gate tunability, which lies at the heart of modern electronics, and, more importantly, is crucial to verify the negative longitudinal MR as a signature of topological semimetal. The negative longitudinal MR in topological semimetals arises from their 'monopoles' in momentum space, which generate a non-trivial Berry curvature that couples an external magnetic field to the velocity of electrons. As a result, an extra chiral current can be induced in parallel magnetic fields, leading to the negative MR. As the Berry curvature diverges at the Weyl nodes[23,25], the negative longitudinal MR is expected to be maximized at the Weyl nodes. Therefore, to verify the negative longitudinal MR as a signature of topological semimetal, it is crucial to measure its dependence on the carrier density with a tunable gate voltage in situ. So far, no such experiment has been reported in WSMs.

Gate-tunable negative longitudinal MR effect in $WTe_2$ thin films has been observed in most studied devices. Figure 3a shows the longitudinal MR of sample #1 for various back gate voltages $V_{bg}$ from $-40$ to $40$ V. The negative longitudinal MR was pronounced at $-40$ V, it was gradually suppressed as $V_{bg}$ increased and was nearly completely suppressed at $40$ V, as indicated by the extracted $C_W$ plot in Fig. 3b (see fitting results in Supplementary Fig. 7). This result implies that as $V_{bg}$ increases, the Fermi energy increases and moves away from the Weyl points from above. In contrast, an opposite trend (monotonously increasing $C_W$ with increasing $V_{bg}$) was observed in sample #3 (as shown in the inset of Fig. 3b and see Supplementary Fig. 8), suggesting that the Fermi energy approaches the Weyl points from below.

More interestingly, a non-monotonous $C_W - V_{bg}$ curve is observed in sample #2 with $C_W$ maximized at certain $V_{bg}$. As shown in Fig. 3c, as $V_{bg}$ increases from $-40$ to $0$ V, the native longitudinal MR is gradually enhanced until reaching a maximum between $0$ and $20$ V. When higher $V_{bg}$ is applied, the native longitudinal MR is apparently suppressed. The $C_W$ data extracted from the complete data set are plotted in Fig. 3d, showing the maximum value of $C_W$ in the range of $10$–$17.5$ V. As the anomaly conductivity reaches the maximum, while crossing the Weyl points, these results indicate that we can successfully access the Weyl points via gate tuning. While modulating other bulk WSMs is mostly achieved through chemical/physical doping approaches and the material properties are fixed by the selected composition and doping level during material processing, the in-situ tuning of the Fermi energy in layered type-II WSMs could provide an important platform to explore Chiral physics of type-II Weyl fermions.

## Discussion

In conclusion, our observations of the angle-sensitive negative longitudinal MR and the strong planar orientation dependence in thin $WTe_2$ samples reveal important signatures of chiral anomaly in such a predicted type-II WSM. Taking advantage of the thin-film geometry, we successfully demonstrated the in-situ tuning of the Fermi energy through the Weyl points, resolving the tunability of unique transport properties and verifying the negative longitudinal MR as a signature of topological semimetal. Our results suggest that gated thin $WTe_2$ films may constitute a new and ideal platform to control and exploit the unique properties of type-II Weyl fermions (around the Weyl points) using numerous experimental

techniques and pave the way for the implementation of future chiral electronics.

## Methods

**Materials and devices.** The $WTe_2$ thin films were mechanical exfoliated from single crystals (HQ-graphene, Inc.) onto the silicon substrate covered by $285$ nm $SiO_2$. The thickness of the samples was confirmed by using a Bruker Multimode 8 atomic force microscopy. The electrodes ($5$ nm Ag/$40$ nm Au) were patterned using home-made shadow mask method and deposited by standard electron beam evaporation.

**Experimental setup.** The devices were measured in an Oxford cryostat with a magnetic field of up to $12$ T and based temperature of about $1.6$ K. The MR signals were collected by using a low-frequency Lock-in amplifier. A rotary insert (Oxford Instruments) was used to tilt the angle between the magnetic field and current, $\theta$. As the magnitude and direction of the magnetic field is fixed, rotating a device placed on the rotation unit is equivalent to rotating the magnetic field with a fixed device current direction. The rotary insert has precise control on the tilted angle, with error about $\pm 0.05°$.

**Details on theoretical calculation of anisotropic chiral anomaly in $WTe_2$.** Owing to the $C_{2T}$ symmetry, we can get the general form of the Hamiltonian around a Weyl point, while keeping only terms linear with **k**

$$H(\mathbf{k}) = Ak_x + Bk_y + (ak_x + bk_y)\sigma_y + (ck_x + dk_y)\sigma_z + ek_z\sigma_x$$

The energy spectrum of $H(\mathbf{k})$ can be expressed as

$$\varepsilon_\pm(\mathbf{k}) = Ak_x + Bk_y \pm \sqrt{(ak_x + bk_y)^2 + (ck_x + dk_y)^2 + (ek_z)^2}$$

Hence, the kinetic and potential components can be expressed as

$$T(\mathbf{k}) = Ak_x + Bk_y, \quad U(\mathbf{k}) = \sqrt{(ak_x + bk_y)^2 + (ck_x + dk_y)^2 + (ek_z)^2}$$

We can thus define the ratio around the Weyl point

$$R = (T(\mathbf{k}))^2/(U(\mathbf{k}))^2 = \frac{(Ak_x + Bk_y)^2}{(ak_x + bk_y)^2 + (ck_x + dk_y)^2 + (ek_z)^2}$$

As the direction of $R > 1$ permits the existence of chiral anomaly[32], we calculated the values of $R$ along $a$ and $b$ axes. According to the band structure (see Supplementary Fig. 9), for the Weyl points at $E = 52$ meV with respect to the Fermi level, we can get $R = 0.57$ along $a$ direction, whereas $R = 143.68$ along $b$ direction. For the other four Weyl points at $E = 58$ meV, $R = 0.63$ along $a$ direction, whereas $R = 9.3$ along $b$ direction. The calculated results predict the absence of the chiral anomaly along the direction of $a$ axis and the existence of chiral anomaly along the direction of $b$ axis for all Weyl points, which agree well with our observations in experiments.

**Data availability.** The data that support the findings of this study are available from the corresponding author on request.

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

## Acknowledgements

This work was supported in part by the National Key Basic Research Program of China (2015CB921600 and 2013CBA01603), National Key R&D Program 2016YFA0301700, National Natural Science Foundation of China (11374142, 61574076, 11525417, 11374137 and 11574127), Natural Science Foundation of Jiangsu Province (BK20130544, BK20140017 and BK20150055), Specialized Research Fund for the Doctoral Program of Higher Education (20130091120040), China Postdoctoral Science Foundation, Fundamental Research Funds for the Central Universities and Collaborative Innovation Center of Advanced Microstructures.

## Author contributions

F.M. and Y.W. conceived the project and designed the experiments. Y.W., E.L., J.Z., Y.F., M.W. and K.X. performed the device fabrication and electrical measurements. Y.W., F.M., X.W., H. Lu, B.W., E.L., H.Liu, Y.P. and L.Z. conducted the data analysis and interpretation. Y.W., E.L., Z.H. and Z.W. carried out the Raman spectroscopy measurements and analysis. F.M., X.W., Y.W. and H.Liu co-wrote the paper and all authors contributed to discussions about and the preparation of the manuscript.

## Additional information

**Competing financial interests:** The authors declare no competing financial interests.

