## [Peer Review File · Nature Communications]

Reviewers' comments:

Reviewer #1 (Remarks to the Author):

In this manuscript, the authors report on the low-temperature magneto transport studies on exfoliated thin films of WTe₂, a predicted type-II Weyl semimetal candidate, and observe a negative longitudinal magnetoresistance when the electric and magnetic fields are parallel and along the b axis. As the authors discussed, they attribute this phenomenon to the chiral anomaly in the WSM and further extent their experimental evidence by showing the angle sensitive gate dependent magnetoresistance. These results are noteworthy and have the potential to have a big impact in the field, I would recommend publication in Nature Communications.

Still I have a few comments, the authors need to address before this manuscript is accepted.

My fundamental comment is that the authors should clearly discuss why the negative longitudinal magnetoresistance is not observed in the bulk crystal when magnetic field $H \parallel b$ axis and current $I \parallel b$ axis (Ref. 39), while observed in thin films, although the energy bands of thin films in this manuscript are similar to those of bulk crystals.

The error for the angles in determining the a axis and b axis is approximately $\pm 7.5^\circ$, which is even larger than the angle where the negative longitudinal magnetoresistance disappears, and metal electrodes fabricated with a shadow mask method should not be well aligned with the a axis or b axis, therefore the authors should discuss the influence of these factors.

Meanwhile the authors should briefly describe the experimental setup to align the current and magnetic field.

WTe₂ is readily oxidized via environmental exposure, especially for thin films. Disorder may exist in WTe₂ thin film, how the authors tried to minimize degradation. A point that is possibly overlooked by the authors is the negative longitudinal magnetoresistance in disordered system. The author should discuss how to exclude this possible origin.

Have the authors measured the temperature dependence of resistivity and the magnetoresistance when $H \parallel c$ axis of thin films and compared them with the properties of bulk crystal?

Minor points:

The authors should point out in Fig. 2 and 3 that magnetic field $H \parallel b$ axis and current $I \parallel b$ axis. In Fig. 2 and 3, the magnetoresistance is shown with $\Delta\rho$, it should be replaced with ratio.

Reviewer #2 (Remarks to the Author):

The work presents new and original measurements in a topical Weyl semimetal. Although the results still present some questions, it is clear that authors have gone a great way, tuning the magnetoresistive properties by a backgate. The initial idea has been to make a material thin enough that it can be gated, but thick enough that it maintains the interesting properties of the bandstructure, in this case, Weyl fermion dispersion relation. However, there are too many points lacking for publishing this work. The text leaves too many important unsolved questions.

More detailed comparison with theory is totally lacking, which is surprising, given that bandstructure of this material has been treated in length in literature. Also, some of the most interesting questions, like how to SdH oscillations change with the gate are not addressed at all.

- The most problematic question that is unaddressed in this paper is what is the influence of the interface between the flake and the substrate? Could it be that the exotic properties measured are located at the interface? Authors should discuss this with detail, arguing how and why they can eventually eliminate this circumstance.

- It is not clear why thinning down the material would lead to a suppression of the known magnetoresistant properties (non-saturating magnetoresistance). Authors invoke several effects. These include Coulomb interactions among chiral states or the anisotropy of the Fermi surface. But this is not clear, in particular the issue of the Fermi surface and how this might change in the limit of thin samples. Having seen SdH oscillations, authors could, in principle, comment about their Fermi surface and compare with features known in large macroscopic samples.
- Authors cite current-jetting and magnetic effects, saying that these are not applicable. But they do not explain why these effects are not applicable.
- The finding of pronounced, in-plane anisotropy of the observed effect is certainly interesting, but the discussion must be made also using the known bandstructure of the material. How do bands look like in each direction ? Please explain and elaborate.
- Tuning down to the Weyl point requires to make a comparison with theory. Is the observed gate voltage comparable to expectations from bandstructure ?
- How do Landau levels change with the gate ?

The paper might be considered again after a major revision, including clear explanations and a comparison with theory.

Reviewer #3 (Remarks to the Author):

The authors present a compelling evidence, based on transport measurements in thin films, that WTe₂ is a type-II Weyl semimetal. The latter represent a new class of interesting topological materials in which a Weyl cone that describes the low-energy electron excitations has been "tipped" on its side. As a result the Fermi surface in these materials exhibits an unusual topology with potentially interesting physical manifestations. WTe₂ has been theoretically predicted to be a type-II Weyl semimetal and the manuscript provides good evidence that this is indeed so.

The authors report unambiguous observation of negative magnetoresistance which is believed to be a signature of the chiral anomaly, expected to be present in Weyl semimetals (both type-I and type-II). In addition they find "planar anisotropy" of the effect which they claim as evidence for the type-II behavior. The manuscript is clearly written, for the most part, the data appear robust and their interpretation sound. It will make an excellent contribution to the field that is quickly evolving. I recommend that the manuscript be accepted for publication in Nature Communications after the authors corrected two minor issues listed below.

1) On page 2 the authors state that to observe negative magnetoresistance thin flakes of WTe₂ are required. Why is this so? Chiral anomaly is a bulk effect and the associated negative magnetoresistance should be (and generally is in other materials) observable in bulk samples. The authors should elaborate on this point.

2) The key evidence for type-II behavior of WTe₂ is the observation of "the predicted planar dependence of negative longitudinal MR". Since the assignment of WTe₂ to the type-II class of Weyl materials hinges on this, would it be possible to briefly sketch the physical origin of this prediction?

REVIEWERS' COMMENTS:

Reviewer #1 (Remarks to the Author):

The authors successfully addressed the questions raised by the referees, I recommend publication in Nature Communications.

Reviewer #2 (Remarks to the Author):

I have gone through the new text and the answers to the referees. Authors have carefully revised their manuscript, providing new data and answering questions by all referees. I think that their answers are convincing. The observation of gate-tunable flakes of WTe₂ brings about new interesting physics, radically different from the bulk compound.

Response to Reviewer #1

We thank the referee for the enthusiastic comment on the manuscript's novelty, and for the careful reading of the manuscript. We believe that we fully addressed the referee's comments in the revised manuscript.

1. *My fundamental comment is that the authors should clearly discuss why the negative longitudinal magnetoresistance is not observed in the bulk crystal when magnetic field $H \parallel b$ axis and current $I \parallel b$ axis (Ref. 39), while observed in thin films, although the energy bands of thin films in this manuscript are similar to those of bulk crystals.*

We thank the referee for the good suggestion. **In bulk crystals, the large positive longitudinal magnetoresistance (MR, 1200% in Ref. 39) is the key factor that prevents the observation of the negative longitudinal MR.** Such positive longitudinal MR has been observed in many bulk material systems (like bismuth, TaAs, Cd₃As₂ and so on). Its physical mechanism is still not clear, even though there are some theoretical proposals like Coulomb interaction among chiral states (Ref. 26, 39) or the anisotropy of the Fermi surface (Ref. 46). Thus, to further understand why the positive longitudinal MR gets suppressed for thinner samples is theoretically challenging at current stage and requires more future research efforts.

Following the referee's suggestion, in the revised manuscript, we've added some detailed discussions on page 5 (the end of the 1st paragraph).

2. *The error for the angles in determining the a axis and b axis is approximately $\pm 7.5^\circ$, which is even larger than the angle where the negative longitudinal magnetoresistance disappears, and metal electrodes fabricated with a shadow mask method should not be well aligned with the a axis or b axis, therefore the authors should discuss the influence of these factors. Meanwhile the authors should briefly describe the experimental setup to align the current and magnetic field.*

Figure R1: Schematic structure and measurement circuit of the four-probe devices. The angle between the magnetic field and current is defined as θ .

The angle where the negative longitudinal MR disappears is the angle

between \mathbf{I} and \mathbf{B} marked by θ in Fig. R1 (same as Fig. 1c in the main text). For such angle dependent negative longitudinal MR measurements (with main results plotted in Figure 2), the current was fixed along certain lattice orientation. Thus, **this angle θ has NO correlation with the error for the angles of the current direction relative to the a or b axis.**

For random-shaped flakes, the angle error ($\sim\pm 7.5^\circ$) mainly comes from the Raman measurements. However, in most cases during the experiments, the exfoliated thin samples have at least one straight edge due to mechanical anisotropic properties of WTe_2 . We further used Raman to confirm whether such edge belongs to the a or b axis, in which case determining lattice orientation can be very precise. Thus, the main error of the current direction (relative to the a or b axis) comes from the shadow mask alignment (with an error $\sim\pm 3^\circ$) as the referee pointed out. Here we note again that, this error has NO correlation with the angle(θ)-sensitive measurements performed in Fig. 2.

In our experiments, we used a rotary insert (Oxford Instruments) to tilt the angle θ . While the magnitude and direction of the magnetic field is fixed, rotating a device placed on the rotation unit is equivalent to rotating the magnetic field with a fixed device current direction. The rotary insert has precise control on the tilted angle, with error only about $\pm 0.05^\circ$.

As suggested by the referee, we have added these details in the Methods section.

3. *WTe₂ is readily oxidized via environmental exposure, especially for thin films. Disorder may exist in WTe₂ thin film, how the authors tried to minimize degradation. A point that is possibly overlooked by the authors is the negative longitudinal magnetoresistance in disordered system. The author should discuss how to exclude this possible origin.*

Figure R2: Temperature dependent negative longitudinal MR of a typical device.

We thank the referee for the good suggestion. During our experiments, we did notice that WTe_2 flakes, especially those thinner flakes, are readily oxidized via environmental exposure. While certain sample degradation is unavoidable, we have tried many things to minimize it, mainly focusing on how to limit the

exposure time to the atmosphere by introducing nitrogen environment.

We can also exclude the possible influences of sample degradation induced disorders by examining our electronic transport data. To the best of our knowledge, only two mechanisms related to the disorders can lead to the negative longitudinal MR ($\mathbf{B} // \mathbf{I}$): the weak localization and the Chiral anomaly. The chiral anomaly is expected to be enhanced when approaching the monopoles at the Weyl nodes. The weak localization effect is an universal quantum interference phenomenon, which is very sensitive to the temperature, but NOT sensitive to system details, such as the position of the Fermi level and current direction.

We have three strong experimental evidences to exclude the possible origin of weak localization effect:

1) The temperature dependent longitudinal MR results of a typical sample (Fig. R2) show that the weak anti-localization near the zero field disappeared above 8 K. If the negative longitudinal MR originates from weak localization, it should disappear above 8 K as well. In sharp contrast, the feature of negative longitudinal MR persists at temperatures as high as 30 K.

2) The negative longitudinal MR observed in our devices has shown strong dependence on the gate voltage which effectively tunes the Fermi level.

3) The observation of the absence of negative longitudinal MR along certain lattice direction (a axis). This cannot be explained by weak localization effect, but offers a strong evidence of a type-II WSM.

Following the referee's suggestion, we have added more detailed discussions in the revised manuscript on page 6.

4. *Have the authors measured the temperature dependence of resistivity and the magnetoresistance when $H \parallel c$ axis of thin films and compared them with the properties of bulk crystal?*

Yes, we did perform such measurements with vertical magnetic field applied. The data of temperature dependent resistivity and vertical MR (sample #2) are shown below in Fig. R3. The thin films (7-15 nm) present metallic properties at $B=0$ T and undergo a metal-insulator transition while increasing B , similar to what were observed on bulk crystals. The vertical MR is suppressed in thin films comparing to bulk crystals, which may originate from the scattering of the surface induced suppression of carrier mobility in thin films (Ref 43 and arXiv:1606.05756).

Figure R3: Temperature dependent transverse resistivity and vertical MR of sample #2.

Minor points:

The authors should point out in Fig. 2 and 3 that magnetic field $H \parallel b$ axis and current $I \parallel b$ axis.

In Fig. 2 and 3, the magnetoresistance is shown with $\Delta\rho$, it should be replaced with ratio.

Thank you for the careful reading and good suggestions. We have modified them accordingly in the revised manuscript.

Response to Reviewer #2

We thank the referee for the enthusiastic comment on the manuscript's novelty, and for the careful reading of the manuscript. We believe that we have fully addressed the referee's comments in the revised manuscript.

1. *The most problematic question that is unaddressed in this paper is what is the influence of the interface between the flake and the substrate? Could it be that the exotic properties measured are located at the interface? Authors should discuss this with detail, arguing how and why they can eventually eliminate this circumstance.*

Figure R4: Left: only positive longitudinal MR observed in a thinner sample (5 nm with $\mathbf{B} // \mathbf{I} // b$) at low temperature. Right: temperature dependent negative longitudinal MR (with $\mathbf{B} // \mathbf{I} // b$) of a typical device.

We thank the referee for raising a good point. Indeed, when the layered materials are thinned down, the unneglectable scattering of the substrate could reduce the mobility of the thin films and induce some localization effect (Ref. 43). To clarify this, we performed two additional experiments, with results offering strong evidences to exclude the influence of interface effects on our observations:

- 1) The interface effects should become more predominant for even thinner flakes. To examine this, we performed additional longitudinal MR measurements on thinner films (with thickness thinner than 7 nm). As shown in Fig. R4 (left), the longitudinal MR of a typical device (5 nm) shows positive signal only. This suggests that the negative longitudinal MR observed in our experiments is an intrinsic property of the samples, rather than the interface effects.
- 2) Generally, such localization effects are very sensitive to the temperature, but not sensitive to material details, such as the position of the Fermi level and lattice direction. The temperature dependent longitudinal MR results of a typical sample (Fig. R4 right) show that the feature of negative longitudinal MR persists at temperatures as high as 30 K, while the weak anti-localization near the zero field disappears around 8 K. In addition, the negative longitudinal MR in our samples can be tuned by the back gate voltage and is absent along

certain axis (a axis). These observations cannot be explained by interface localization effects, but offer strong evidences of a type-II WSM.

Following the referee's suggestion, we have added more detailed discussions in the revised manuscript on page 6.

2. *It is not clear why thinning down the material would lead to a suppression of the known magnetoresistant properties (non-saturating magnetoresistance). Authors invoke several effects. These include Coulomb interactions among chiral states or the anisotropy of the Fermi surface. But this is not clear, in particular the issue of the Fermi surface and how this might change in the limit of thin samples. Having seen SdH oscillations, authors could, in principle, comment about their Fermi surface and compare with features known in large macroscopic samples.*

We totally agree with the referee that it is not clear why thinning down the material would lead to a suppression of the known MR properties. In the previous manuscript, we did mention that several mechanisms could induce the positive longitudinal MR (\mathbf{B}/\mathbf{I}), which has been observed in many materials, including bismuth, TaAs, Cd₃As₂ and bulk WTe₂ (~1200%). There are some theoretical proposals like Coulomb interaction among chiral states (Ref. 26, Ref. 39) and the anisotropy of the Fermi surface (Ref 46) trying to explain it, but the physical mechanism is still not clear. Thus, as pointed out by the referee, to understand why the positive longitudinal MR gets suppressed for thinner samples is theoretically challenging at current stage and requires further research efforts. We have added some detailed discussions on page 5 (the end of the 1st paragraph) in the revised manuscript to clarify this point.

3. *Authors cite current-jetting and magnetic effects, saying that these are not applicable. But they do not explain why these effects are not applicable.*

We thank the referee for the good suggestion. In the revised manuscript (page 6), we have explained why these two effects are not applicable in details. The reasons are following:

Since WTe₂ is not a magnetic material, we can safely exclude the possible origin of magnetic effects.

Regarding the current jetting effect, it is usually induced by inhomogeneous currents generated when attaching point contact electrodes (such as using silver paste) to a large bulk crystal. This is not the case for our experimental setup. We used nanofabrication techniques to pattern well-defined electrodes with relatively large electrode-sample contact area to efficiently suppress such effect. And also, as discussed in arXiv:1606.03389 and Ref. 47, the current jetting effect can be suppressed in thin-film samples by comparing to bulk crystals. The negative longitudinal MR observed in our experiments only appeared in thin-film flakes with the thickness ranging from 7 to 15 nm, but not bulk crystals, providing a strong evidence to exclude such current jetting effect.

4. The finding of pronounced, in-plane anisotropy of the observed effect is certainly interesting, but the discussion must be made also using the known bandstructure of the material. How do bands look like in each direction? Please explain and elaborate.

We thank the referee for the good suggestion to elaborate on this point. Following this suggestion, we have added the band structure and calculation results in the supplementary materials.

Figure R5 : Band structure results of WTe₂ along the Y-Γ-X direction with SOC.

As presented in Nature 527, 495-498 (2015): “The chiral anomaly appears in a WP2 only when the direction of the magnetic field is within a cone where $|T(\mathbf{k})| > |U(\mathbf{k})|$. If the field direction is outside this cone, then the Landau-level spectrum is gapped and has no chiral zero mode.”

Owing to the C_{2T} symmetry, we can get the general form of the Hamiltonian around a Weyl point while keeping only terms linear with \mathbf{k}

$$H(\mathbf{k}) = Ak_x + Bk_y + (ak_x + bk_y)\sigma_y + (ck_x + dk_y)\sigma_z + ek_z\sigma_x$$

The energy spectrum of $H(\mathbf{k})$ can be expressed as

$$\varepsilon_{\pm}(\mathbf{k}) = Ak_x + Bk_y \pm \sqrt{(ak_x + bk_y)^2 + (ck_x + dk_y)^2 + (ek_z)^2}$$

Hence the kinetic and potential components can be expressed as

$$T(\mathbf{k}) = Ak_x + Bk_y, U(\mathbf{k}) = \sqrt{(ak_x + bk_y)^2 + (ck_x + dk_y)^2 + (ek_z)^2}$$

We can thus define the ratio around the Weyl point

$$R = (T(\mathbf{k}))^2 / (U(\mathbf{k}))^2 = \frac{(Ak_x + Bk_y)^2}{(ak_x + bk_y)^2 + (ck_x + dk_y)^2 + (ek_z)^2}$$

While the direction of $R > 1$ permits the existence of chiral anomaly, we calculated the values of R along \mathbf{a} and \mathbf{b} axes. For the Weyl points at $E = 52$ meV with respect to the Fermi level, we can get $R = 0.57$ along \mathbf{a} direction while

R=143.68 along \mathbf{b} direction. For the other four Weyl points at E=58 meV, R=0.63 along \mathbf{a} direction while R=9.3 along \mathbf{b} direction. The calculated results predict the absence of the chiral anomaly along the direction of \mathbf{a} axis, and the existence of chiral anomaly along the direction of \mathbf{b} axis for all Weyl points, which agree well with our observations in experiments.

5. *Tuning down to the Weyl point requires to make a comparison with theory. Is the observed gate voltage comparable to expectations from bandstructure?*

We thank the referee for the good suggestion. While it is challenging to determine the Fermi energy of the measure samples due to complicated fermi surface of WTe₂, we calculated the carrier density required to tune the Fermi energy cross the two Weyl nodes by using DFT calculations. By comparing it with the carrier doping tunability of the applied back gate, together with the experimental evidence of the gate-tunable negative longitudinal MR, we believe that we did access the Weyl nodes via the electric field effect.

We first performed the electronic band structure calculations using VASP packages. The carrier density to tune the Fermi energy cross the two Weyl nodes (52 meV to 58 meV) was calculated to be about $3.5 \times 10^{12} \text{ cm}^{-2}$ through the integration of density of states.

Regarding the doping tunability of gate voltage V_g , the carrier density n can be calculated by:

$$n = \frac{\epsilon_r \epsilon_0 V_g}{te}$$

where $\epsilon_r = 3.9$ is the dielectric constant (or called the relative permittivity) of SiO₂, $\epsilon_0 \approx 8.85 \times 10^{-12} \text{ F/m}$ is the vacuum permittivity, $t=285 \text{ nm}$ is the thickness of SiO₂ dielectric, e is the static electron charge. Numerically, $n/V_g \approx 7.6 \times 10^{10} \text{ cm}^{-2}$. During our experiments, the gate voltage V_g was tuned from 40 V to -40 V, resulting a tuned carrier density about $6.1 \times 10^{12} \text{ cm}^{-2}$. This value is larger than the minimum required value ($\sim 3.5 \times 10^{12} \text{ cm}^{-2}$) based on the above theoretical calculation, suggesting the Weyl points can be accessed via the effect field effect in our experimental setup.

6. How do Landau levels change with the gate?

Figure R6: Back gate dependence of SDH oscillations observed in a typical device (sample #1).

We measured the SDH oscillation of a typical device (sample #1) at different gate voltages, with data shown in Fig. R6. The shift of the oscillation was observed, indicating successful tuning of Fermi energy relative to the Landau levels.

Response to Reviewer #3

We thank the referee for the careful reading and the enthusiastic comment on the manuscript's importance. We believe that we fully addressed the referee's comments in the revised manuscript.

1. *On page 2 the authors state that to observe negative magnetoresistance thin flakes of WTe₂ are required. Why is this so? Chiral anomaly is a bulk effect and the associated negative magnetoresistance should be (and generally is in other materials) observable in bulk samples. The authors should elaborate on this point.*

We thank the referee for the good suggestion. In bulk crystals, the large positive longitudinal magnetoresistance (MR, 1200% in Ref. 39) is the key factor that prevents the observation of the negative longitudinal MR. Such positive longitudinal MR has been observed in many bulk material systems (like bismuth, TaAs, Cd₃As₂ and so on). Its physical mechanism is still not clear, even though there are some theoretical proposals like Coulomb interaction among chiral states (Ref. 26, 39) or the anisotropy of the Fermi surface (Ref. 46). Thus, to further understand why the positive longitudinal MR gets suppressed for thinner samples is theoretically challenging at current stage and requires more future research efforts.

Following the referee's suggestion, in the revised manuscript, we've added some detailed discussions on page 5 (the end of the 1st paragraph).

2. *The key evidence for type-II behavior of WTe₂ is the observation of "the predicted planar dependence of negative longitudinal MR". Since the assignment of WTe₂ to the type-II class of Weyl materials hinges on this, would it be possible to briefly sketch the physical origin of this prediction?*

We thank the referee for the good suggestion to sketch the physical origin of this prediction. Following this suggestion, we have added the detailed discussions in the supplementary materials.

In details, as presented in Nature 527, 495-498 (2015): "The chiral anomaly appears in a WP2 only when the direction of the magnetic field is within a cone where $|T(\mathbf{k})| > |U(\mathbf{k})|$. If the field direction is outside this cone, then the Landau-level spectrum is gapped and has no chiral zero mode."

Owing to the C_{2T} symmetry, we can get the general form of the Hamiltonian around a Weyl point while keeping only terms linear with \mathbf{k}

$$H(\mathbf{k}) = Ak_x + Bk_y + (ak_x + bk_y)\sigma_y + (ck_x + dk_y)\sigma_z + ek_z\sigma_x$$

The energy spectrum of $H(\mathbf{k})$ can be expressed as

$$\varepsilon_{\pm}(\mathbf{k}) = Ak_x + Bk_y \pm \sqrt{(ak_x + bk_y)^2 + (ck_x + dk_y)^2 + (ek_z)^2}$$

Hence the kinetic and potential components can be expressed as

$$T(\mathbf{k}) = Ak_x + Bk_y, U(\mathbf{k}) = \sqrt{(ak_x + bk_y)^2 + (ck_x + dk_y)^2 + (ek_z)^2}$$

We can thus define the ratio around the Weyl point

$$R = (T(\mathbf{k}))^2 / (U(\mathbf{k}))^2 = \frac{(Ak_x + Bk_y)^2}{(ak_x + bk_y)^2 + (ck_x + dk_y)^2 + (ek_z)^2}$$

While the direction of $R > 1$ permits the existence of chiral anomaly, we calculated the values of R along \mathbf{a} and \mathbf{b} axes. For the Weyl points at $E=52$ meV with respect to the Fermi level, we can get $R=0.57$ along \mathbf{a} direction while $R=143.68$ along \mathbf{b} direction. For the other four Weyl points at $E=58$ meV, $R=0.63$ along \mathbf{a} direction while $R=9.3$ along \mathbf{b} direction. The calculated results predict the absence of the chiral anomaly along the direction of \mathbf{a} axis, and the existence of chiral anomaly along the direction of \mathbf{b} axis for all Weyl points, which agree well with our observations in experiments.